



# SaLEM (v1.0) - A Soil and Landscape Evolution Model for simulation of regolith depth in periglacial environments

Michael Bock[1], Olaf Conrad[1], Andreas Günther[2], Ernst Gehrt[3], Rainer Baritz[2], and Jürgen Böhner[1]

[1]Institute for Geography, University of Hamburg, Germany
[2]Federal Institute for Geosciences and Natural Resources (BGR), Hannover, Germany
[3]State Authority for Mining, Energy and Geology (LBEG), Hannover, Germany

*Correspondence to*: Michael Bock (michael.bock@uni-hamburg.de)

**Abstract.** We propose the implementation of a soil and landscape evolution model (SaLEM) for the spatiotemporal investigation of soil parent material evolution following a lithologically differentiated approach. Relevant parts of the established model GOLEM have been adapted for an operational GIS tool within the open source software framework SAGA, thus taking advantage of SAGA's capabilities for geomorphometric analyses. The model is driven by paleo-climatic data (temperature, precipitation) representative for periglacial areas in Northern Germany over the last 50.000 years. The initial conditions have been determined for a test site by a digital terrain model and a geological model. Weathering, erosion and transport functions are calibrated using extrinsic (climatic) and intrinsic (lithologic) parameter data. First results indicate that our differentiated SaLEM approach shows some evidence for the spatiotemporal prediction of important soil parental material properties particularly its depth. Future research will focus on the validation of the results against field data, and the influence of discrete events (mass movements, floods) on soil parent material formation has to be evaluated.

## 1 Introduction

The properties of nowadays soils rely to a large extent on their development under past climatic conditions. Particular in areas of the world, where periglacial conditions ruled the soil forming processes during the pleistocene our understanding of soils could improve substantially, if more reliable information about their historical formation would be available. Thus we introduce an operational tool designed for the spatiotemporal prediction of parent material depth of soil formations utilizing a landscape evolution model (LEM). The model has been developed to operate in a GIS environment allowing for lithologically differentiated surface process simulations. More specifically, it has been designed for the modelling of the spatial distribution and properties of periglacial sediments and regolith formation processes in central European mountainous areas that were unglaciated during the Pleistocene, and is currently solely applicable to this particular geoenvironmental setting.

The model has been implemented within the framework of SAGA, System for Automated Geoscientific Analyses, which is an open source Geographical Information System (GIS) platform (Conrad, 2007, Conrad et al., 2015). To emphasize its focus on soil formation processes we call it Soil and Landscape Evolution Model (SaLEM). Compared to GOLEM





(Geomorphic/Orogenic Landscape Evolution Model; Tucker & Slingerland, 1997), which has been chosen as a starting point for our own developments, SaLEM represents a very specialized type of LEM in terms of time scale, spatial domain and landscape forming processes. With respect to soil forming processes the original GOLEM code was substantially revised, transferred and expanded with the permission of the author into the SAGA environment. GOLEM's original intention was to

model the interaction between landscape evolution and geodynamic processes over longer geologic time periods for large areas. In turn, SaLEM aims to model the formation of weathering layers in lithologically differentiated terrains interacting between processes as erosion, transport and sedimentation that all together govern the development of soil parent material over the last several ten thousand years.

The essential significance of the geological parent material for the general formation of soils is widely recognized since the

first half of the last century. Jenny (1941) was the first to formulate a functional relationship between important soil parameters and various local factors, such as the climate, organisms, topography, time and parent material in his famous soil equation. Even though this functional relationship was not expressed numerically; the theoretical considerations of Jenny (1941) are the basis of today's process-oriented modeling in soil sciences. Digital soil mapping (see Lagacherie, et al. 2007, McBratney et al., 2003, Behrens & Scholten, 2006) developed mostly statistical and geostatistical models to indirectly

predict specific physical or chemical properties of soils incorporating specific spatial uncertainties. However, the majority of these approaches are not process based, therefore being capable for site-specific soil property data regionalization but do not contribute to the understanding of the factor correlations.

In contrast, deductive models represent dynamic processes by mapping the functional relationship of the sub-processes and thus can contribute in addition to data delivery to the particular process understanding (Böhner, 2006). In this sense, the

process-oriented SaLEM tries to model the parent material of soils for natural environments.

Data on soil parental material consisting either of in-situ weathered rocks, weathered loose sediments or even weathered paleosoils are highly underrepresented in geoscientific data sets. While geological maps in mountainous terrains mostly display petrographical and stratigraphical properties of solid (unweathered) bedrock and loose quaternary sediments are often underrepresented and undifferentiated, soil maps indicate the spatial distribution of soil types which also do not allow

for spatial identification of regolithic or sedimentary features. This gap between spatial distributed data to soils and bedrock is significant in almost all data held by Geological Surveys. Filling this gap has been perceived as important nowadays because this critical zone has been recognized as the place where the "Earth's weathering engine provides nutrients to nourish ecosystems and human society, controls water runoff and infiltration, mediates the release and transport of toxins to the biosphere, and conduits for the water that erodes bedrock." (Brantley et al., 2006, p 4)

Holocene soil formation, however, takes place on exactly this material and therefore adapts the material properties of the regolith as for example grain size composition, bulk density, mineral composition´, porosity, permeability, etc., that all depend directly on the physical properties of the parent material.



In most cases, the weathered part of the geological substratum on which soil develops is considerably thicker than the soil itself. For water balance models, simulations for migration and filtering of pollutants, shallow groundwater flow modeling or erosion and terrain stability modeling, information on physical and chemical properties of the total regolith is mandatory.

We describe the background of SaLEM and the state of its development. Special emphasis is given to the site-specific

modeling of regolith depth and sediment formations in a periglacial geo-environmental setting as this is highly influenced by the supply of allochthonous, aeolian sediments (loess). We discuss the climatic factors driving soil and landscape evolution in north-central Germany during the Pleistocene. We suggest a parameterization for weathering rates. The final model has been applied and evaluated in a case study for a pilot area in northwest Germany. The results show that there is a need to improve the spatiotemporal identification and quantification of regolith forming processes, and the prediction of first-order

geomechanical and chemical properties of parent material of soils.

## 2 Study area

The study area Ebergoetzen is part of the German low mountain range, which is bordered to the north by a major continental fault system ("Elbe fault system, Fig. 1b). This mountainous area was free of ice during the Younger Pleistocene (Fig. 1a), but it was exposed to periglacial climatic conditions. To the north it is adjacent to the glacier-formed North German

Lowland.



**Figure 1. Location of the study site: a) Glacial ice sheets of LGM in Europe, data from Ehlers & Gibbard, 2004, b) Loess deposits in Germany, data from Haase et al., 2007, test site 'Ebergoetzen' as black rectangle, c) Simplified Geological map of test site 'Ebergoetzen', according to Ehlbracht, 2000. For our purpose the areas with quaternary deposits were removed.**



The study area is geomorphologically characterized by escarpments formed by Triassic sedimentary rocks of the Germanic Basin. The north German escarpment setting is shaped by NNE-SSW striking major fault zones of paleozoic (Variscan) origin that were reactivated as sinistral transcurrent fault systems during Mesozoic (Alpidic) deformations (Mazur & Scheck-Wenderoth, 2005, Fig. 1b). Mesozoic transtensional deformations accompanied by salt tectonics led to development of half

graben structures and tilting of discrete upper crustal segments forming escarpments.

Specifically, the study area 'Ebergoetzen' (location in Fig. 1b, simplified Geological Map (Ehlbracht, 2000, Fig. 1c) is formed by two escarpments with corresponding flats and slopes. Roughly speaking, this can be described as follows:

•        The western part of the area is dominated by a gently westward inclined surface built of Triassic limestones of the "Lower Muschelkalk" in relatively high altitudes (about 420m above sea level). The "Lower Muschelkalk" is underlain by

Lower Triassic claystones and siltstones of the "Upper Buntsandstein" that forms the escarpment of the "Göttinger Wald".

•        To the East, a slightly westward inclined surface (elevation about 290m above sea level) consisting of red sandstones of the "Middle Buntsandstein 2" is exposed. This surface is bordered by a steeply sloping section at the base of the escarpment which is made of sequences of sand- and siltstones of the "Middle Buntsandstein 1".

•        Further to the East, in general the importance and thickness of loess rises. Partly as insular very thick resources can

be found (> 10 meters).

During the glaciated periods of the Pleistocene, the periglacial environment of our study site was characterized by intensive weathering, erosion and transport processes. Frost weathering of numerous freeze-thaw-cycles resulted in loosening of the exposed sedimentary bedrock mainly along joint surfaces. Additionally, extensive dissolution of the calcareous rocks of the "Muschelkalk" fragmented these lithological successions. The crushed rock was released from the rock mass and -

dependent on the local terrain situation - remained in situ or was moved downhill by solifluction, creeping and mass wasting processes. The intensity of the weathering processes as well as the speed of the transport processes depends on the material properties of the rock resp. rock debris but is also altered by allochthonous input of loess material. Figure 1b shows that the spatial distribution of loess deposits of distinct thickness is a general phenomenon in wide areas of Germany.

## 3 Materials and Methods

The general purpose of LEM is a better understanding of landscape history through a simulation of landforming processes and process interactions (Tucker & Hancock, 2010). The main purpose of SaLEM is the mapping of regolith properties according to known physical relationships. In absence of reliable data for certain process variables these have to be substituted by suitable parameterizations.

### 3.1 Methodological Background

SaLEM has been developed using the SAGA (System for Automated Geoscientific Analyses) framework, which is an open source software that provides an extensive application programming interface dedicated to spatial data analysis and



visualization (Conrad et al., 2015). SaLEM simulates the dynamics of selected landscape-forming processes (weathering, erosion, transport and deposition), thus representing an operational GIS tool for numerical process modeling. Differential equations used in the model are based on simplified physical models, such as the description of weathering or transport processes. The original C-code of GOLEM (Tucker & Slingerland, 1997) was ported to the C++ based environment of

SAGA. Tucker & Slingerland describe the aim of GOLEM as the exploration of the interaction of tectonics (uplift) and erosion for the landscape over long geological timescales (several Ma). The goal of SaLEM is the lithologically differentiated modeling of weathering, erosion, transport and deposition of unconsolidated material covering the bedrock for comparably shorter periods (recent 50-100 ka). The part of GOLEM that in particular is relevant for these objectives, is the sub-model for diffusive regolith creep. With the focus on the prediction of parent material for soil formation it does not

consider landscape compartments that are beyond this scope. Accordingly, the modeling of fluvial incision and transport as tectonic uplift was not adopted. GOLEM's function for regolith production (or weathering) was replaced by a set of rock specific and climate sensitive equations considering frost and chemical weathering separately. The simulation time is free selectable and depends only on the availability of climate data, which are necessary to drive the model.

One problem for LEM based forward modeling is the impossibility to reconstruct the initial paleo-topographic situation. This

problem is known as equifinality or convergence of landforms and was discussed many times in geomorphographic papers (e.g. Odoni, 2007, Peeters et al., 2006). It must be considered highly evident when modeling over longer geological time spans (several Ma), however for the time frame considered here (50 ka) it can be proposed as less important (Peeters et al., 2006). Therefore, we use the actual topography as predefined by the DEM as the initial topography for our modeling.

The layer of unconsolidated material, which today can be found omnipresent covering the bedrock is the result of many

natural processes that interacted for many thousands of years. Solid bedrock is attacked by two categories of weathering processes: loosening of the rock mass by physical weathering, and rebuilding of the mineral constituents by chemical weathering. When individual fragments are separated from the bedrock, the unconsolidated material (regolith) is exposed to downhill transportation by gravitational processes. During the Pleistocene, discrete episodes of intense mixing of the unconsolidated layer with allochthonous materials are evident, in particular the aeolian sediments known as loess. This took

place under the influence of vegetation and resulted in a multi-material-layer covering the solid rocks of the mountainous areas with a thin coat of regolith. The thickness of this coat may range from a few centimeters up to several meters. Due to its physico-chemical properties its proportion in regolith influences current soil properties significantly.

Basically all weathering and transport-related processes follow physical and chemical laws that should be reflected by the model. However, this can only be done in an approximation to the real world phenomena due to several reasons: Input data

are not available to all factors of the involved processes, the spatial resolution is not applicable to model all processes realistically, and still physical modeling of some of the involved processes would be too complex and beyond the scope of SaLEM. Thus modeling is limited to processes that can be depicted and empirically described. This general feature of reduction becomes especially clear in the case of modeling the periglacial layer as parent material, because many processes, such as the influence of vegetation on erosion, transport and allochthonous deposits, remain unconsidered.





## 3.2 Climate data

The climatic development of the northern hemisphere during the Pleistocene is fairly well known nowadays due to recent methodological developments in paleoclimatology. Through the introduction of ice core analysis as proxies it has become possible to reconstruct the course of long time series of climatic elements, although the derived information applies only to

the locations where the data is taken from (Bubenzer & Radtke, 2007). Palaeo-climate modelling data of world records is now available in relatively high spatial and temporal resolution.

For the calibration of chemical and physical (frost) weathering two climate data sets are considered, one for the long term temperature signal and one as scenario representing the annual/seasonal climate.

The long term signal has been taken from the ice core project GISP2 (Alley, 2000). It provides 30 years temperature

averages for the last 50.000 years. These temperatures have been calibrated to the annual mean temperatures of our study site. Figure 2 shows the course of temperature, which was derived for the location from the O16/O18 isotope ratio.

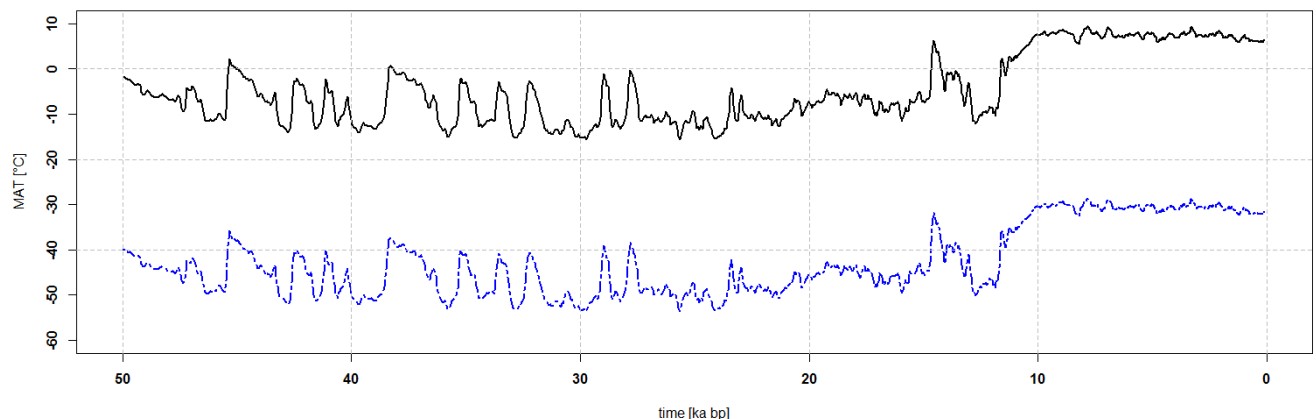

**Figure 2 Derived mean annual temperature (MAT) data for the GISP2 location (dashed line) and the assumed curve for the test site 'Ebergoetzen'(solid line) (after Alley, 2000)**

1631 values are available for a period of 50,000 years, which means an average of 30 years resulting in one mean temperature value. Although these are unevenly distributed: 17 values for the most past 1000 years, 115 values for the most recent 100 years. The average shown in the data is still more accurate than the required temporal resolution of the model. The curve ends up with the value of minus 31°C as the current mean annual temperature of the GISP2 location. SaLEM raises the entire curve to the actual mean annual temperature level of the respective working area via the user interface.

For the annual variation of the temperature signal, a temporal resolution of 6 hours and spatial resolution of about 210 km temperature- and precipitation data was extracted from the global NCEP/NCAR reanalysis programme covering the last 40 years (Kalnay et al., 1996). From this data set a time series of a recent periglacial environment (Timan Ridge, Russia) has been chosen to act as analogue for the annual pleistocene temperature and precipitation pattern at our study site (Fig. 3). The



average of nine annual variations of the NCEP/NCAR data was then referenced to each temperature datum of the calibrated GISP2 curve.

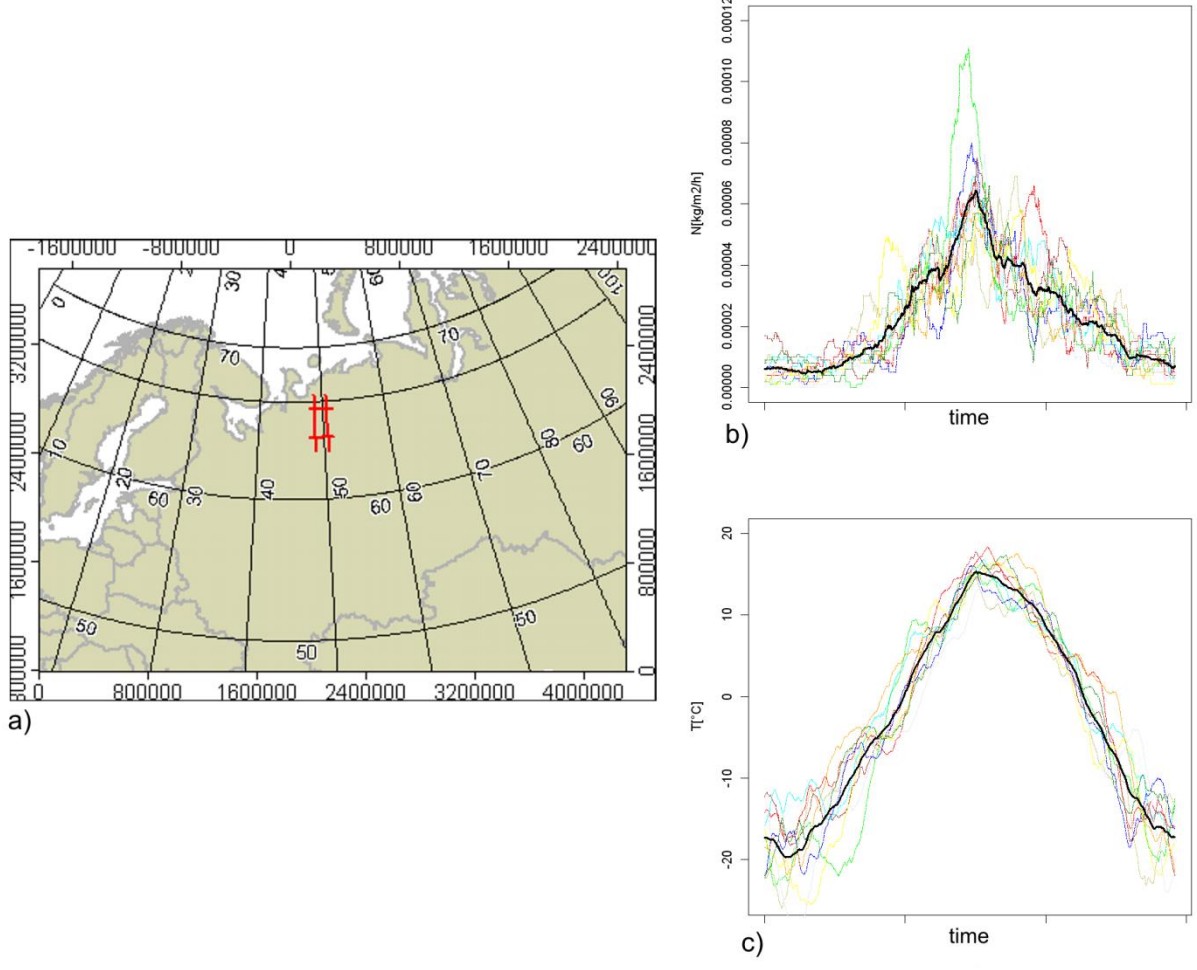

**Figure 3. Location (3a) and annual variations of temperature (3b) and precipitation (3c) for 9 years of the Timan Ridge, Russia, derived from NCEP/NCAR Reanalysis data, Kalnay et al., 1996. The model uses the average curves (bold black line).**

Both the GISP2 data for paleo-temperatures and the NCEP/NCAR reanalysis data including the annual variations of precipitation and temperatures are provided to the user of SaLEM. Via a temperature offset, the level of the GISP2 curve can be moved up or down to calibrate it to different sites.

### 3.3 Bedrock geology and weathering indices

SaLEM operates on a geological model consisting of elevation-registered grids representing lithological contacts and topography (DEM). For simplification, the model uses the current topography represented by a DEM (50m spatial



resolution) as the initial starting point. For our study, a geological subsurface model was constructed from geological map information (Ehlbracht, 2000), two geological cross-sections, a deep borehole and DEM data (Fig. 4b). For model construction, first the outcrop lines of the geological units were elevation-registered with the DEM data, and the geological cross-sections were vectorized and transferred into 3-D space (Fig. 4a). Subsequently, geological surfaces were constructed

with the outcrop line and cross-section line data using the geomodeller GOCAD® (Paradigm, 2015). Last, thickness raster data for each lithological unit were calculated on the same resolution as the DEM data and assigned to each geological unit. This data then serves as geometrical lithological input information for SaLEM.

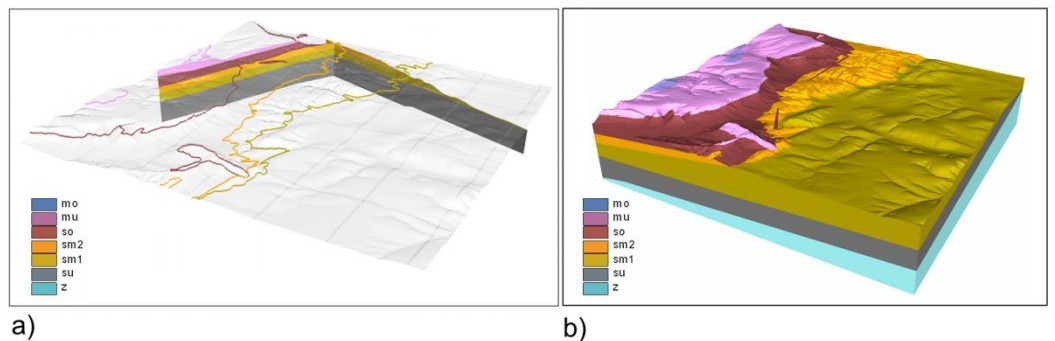

**Figure 4. Geological units in test site 'Ebergoetzen', elevation-registered with the DEM data (4b) and the geological cross-sections**
**(4a) derived from Ehlbracht, 2000. Upper Muschelkalk (mo), Lower Muschelkalk (mu), Upper Buntsandstein (so), Middle Buntsandstein2 (sm2), Middle Buntsandstein1 (sm1), Lower Buntsandstein (su), Zechstein (z)**

The weathering susceptibility of the different lithological model units was assigned through expert-derived chemical- and physical weathering indices as proposed by Gehrt (2008), for the lithological successions of Northern Germany. Gehrt (2008), arranged the 75 stratigraphic units occurring in Lower Saxony regarding their resistance against weathering of their

rock types at an ordinal scale (1: very resistant to 5: least resistant). Since the indexes are not calculated from measured data, only the relative differences of the different rock types are used here. From this knowledge, the weathering equations adapted from Temme & Veldkamp (2009), were calibrated for each model time step to obtain the weathering rate through equations like the well known "humped model" (for chemical weathering) inter alia.

The applied weathering equations go back to Bloom (1998) (1) resp Cox (1980) (2):

$$F_0 * \frac{(T + (a * R)) - T_{max}}{(T_{min} - T_{max}) * \cos \beta}$$
(1)

Equation 1 for frost weathering in mm year[-1], where $F_0$ is the maximum frost weathering on a flat surface, $\alpha$ is the buffering parameter for thickness of the regolith layer, R the thickness of the regolith layer), $\cos \beta$ is the cosine of slope

$$-(P_0(e^{-k_1} - e^{-k_2}) + P_a)$$
(2)



Equation 2 for chemical weathering in mm year$^{-1}$, where $P_0$ is the maximum chemical weathering rate, $P_a$ is the chemical weathering in steady state, $k_1$ is the weathering rate constant before the maximum rate is reached. With further increasing regolith thickness the rate of chemical weathering decreases again, $k_2$ is the weathering rate constant after the maximum rate.)

$F_0$, a, $P_a$, $k_1$ and $k_2$ are constants which are dependent on the material. In a lithological differentiated approach like SaLEM the values for these constants were changed relative to each other according to Gehrt (2008) (see table 1):

**Table 1: Resistance against weathering (frost weathering and chemical weathering) of different triassic bedrock types occurring in site 'Ebergoetzen' after Gehrt (2008) and derived initial values for the parameters of SaLEMs weathering equations (1) and (2)**

|  | Upper Muschelkalk | Lower Muschelkalk | Upper Buntsandstein | Middle Buntsandstein 2 | Middle Buntsandstein 1 |
|---|---|---|---|---|---|
| Resistance after Gehrt (2008) | 1 | 3 | 5 | 5 | 5 |
| $F_0$ | 0.002 | 0.005 | 0.010 | 0.010 | 0.010 |
| a | 0.0010 | 0.0015 | 0.003 | 0.003 | 0.003 |
| $P_a$ | 0.0006 | 0.0008 | 0.0020 | 0.0020 | 0.002 |
| $k_1$ | 4 | 4 | 4 | 4 | 4 |
| $k_2$ | 6 | 6 | 6 | 6 | 6 |

**3.4 Allochthonous deposits**

One formative phenomenon of the periglacial deposits in Central Europe is their partly large proportion of not in-situ produced materials. These are designated as "allochthonous" materials consisting of the terrestrial, aeolian sediment loess.

In the absence of real measurement data describing spatially distributed loess deposition rates, a simple model was developed to indicate loess accumulation rates per year for each grid cell. These rates were derived from work done by

Frechen et al. (2003), who calculated accumulation rates from loess profiles all over Central Europe. The rates determined by Frechen et al. (2003) differ from 100 to more than 7000 g m$^{-2}$ year$^{-1}$ for a period from 28 – 18 ka BP resp 300 to more than 4000 g m$^{-2}$ year$^{-1}$ for a period 13 – 18 ka BP. To apply the discrete accumulation rates to the spatial SaLEM context, the SAGA module Wind Effect (Windward / Leeward Index, Böhner & Antonic, 2008) is parameterized on the basis of windward and leeward effects derived from a DEM taking into account a prevailing wind direction. In other words the relief

information is recalculated to index values dependent on the exposure to the assumed wind direction. As a prevailing wind direction during LGM in Central Europe the direction was set to ENE going back to Roche et al. (2007). The literature values for loess accumulation by Frechen et al. (2003) were translated in thickness per grid cell and stretched on the result of the index calculation (Fig. 5).



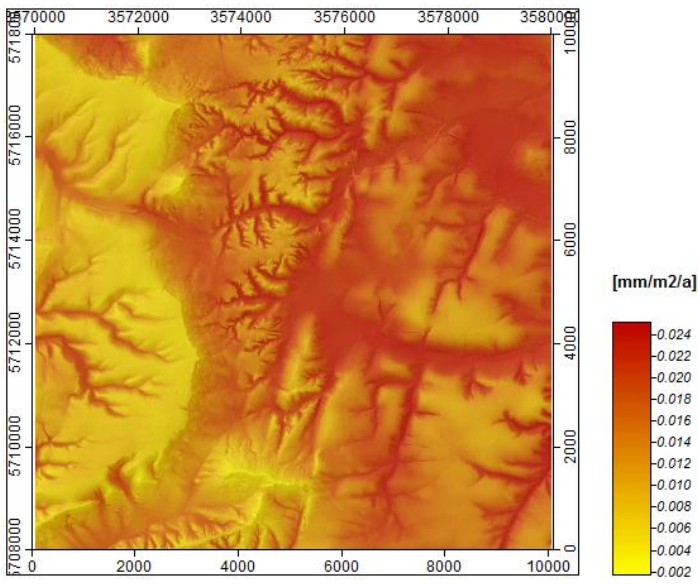

**Figure 5 Parameterization of loess accumulation rate for Ebergoetzen: DEM derived parameter windward/leeward effect (Böhner & Antonic, 2008) combined with mass accumulation rates after Frechen et al. (2003) for period 28-18 ka BP.**

For each time step in the modelling, the allochthonous input is simulated after the weathering process and before the downhill transport of the material. A spatially differentiated amount of loess material is accumulated on the grid cells. This information is passed to the model for each specific grid cell.

### 3.5 Transport

The simulation of hillslope sediment transport is modelled as a diffusion process, a concept that is commonly used for sediment flux modelling (e.g. Tucker & Slingerland, 1997, Pelletier, 2008, Anderson & Anderson, 2010, Gillespie, 2011). It relates to Fick's law of diffusion and is used to describe the sediment flow in dependence of time and slope gradient and results in a rate of change in elevation, expressed as

$$\frac{\partial h}{\partial t} = k_d \nabla^2 h$$

(3)

where h is the elevation, t is the time, and $k_d$ the hillslope diffusivity coefficient, which determines the speed of the diffusive sediment transport. Because sediment fluxes should be restricted to the unconsolidated regolith cover, the maximum allowed rate of change in elevation has been limited to the regolith thickness. Here SaLEM closely follows the original GOLEM implementation.

While the quantification of sediment transport and its associated denudation and deposition follows a well established approach, it does not give information about the sediment composition. In order to overcome this restriction we developed a





tracer concept for the model. Such tracers represent soil particles, which are released evenly distributed in the regolith layer. The information that a tracer stores is its geographical position, the depth at which it is buried, and the geological unit from which it was released. The closer a tracer is to the surface, the higher is the probability that it becomes moved by diffusive hillslope transport. The decision if a tracer is moved in a simulation time step is made with a depth dependent random

function. If a tracer is moved it follows the direction of the slope aspect. The covered distance is estimated as a function of slope and hillslope diffusivity coefficient. To reflect uncertainties in the tracer path simulation, a degree of randomness can be added to the direction, distance and depth at which it will be deposited again. For each tracer its path can be stored in an additional data set. Further information can be collected about the time and the duration of its transportation (Fig. 6).

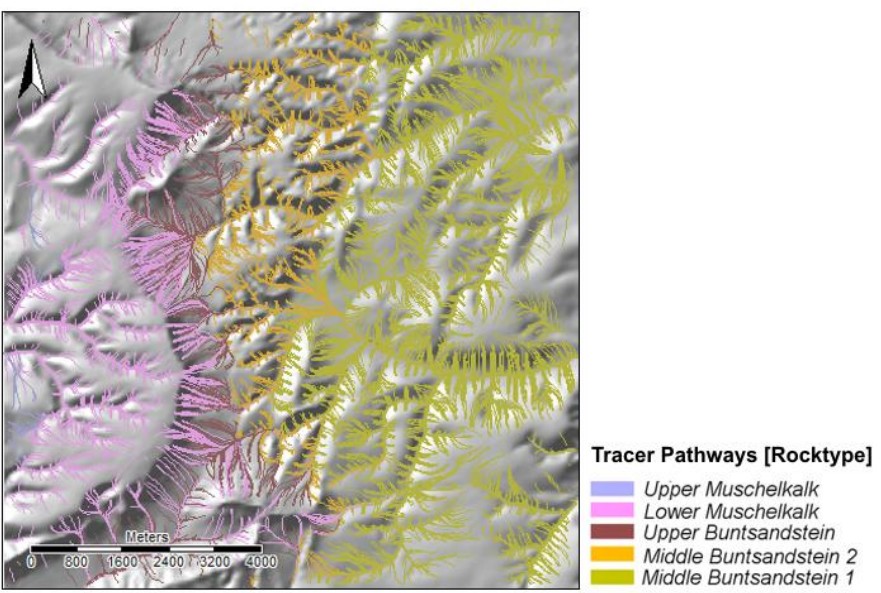

**Figure 6 Transport pathways in Ebergoetzen of the virtual tracer particles from the location of their release from the rock via weathering and erosion to the place where the transport stops.**

**3.6 Model run**

A model run is executed for the specified time range using a discrete time step size, typically 100 years. Initializations done before the model run comprise the loading of the climate data base, the validation of weathering equations and the definition

of an initial regolith cover. Now the same processing scheme is applied for each time step. At first allochthonous input, if specified, is added to the regolith cover. This also increases the surface elevation. Next step is the bed rock weathering, which will increase the regolith cover without changing the surface elevation. The weathering rate depends on regolith thickness, climate variables and rock type specific equations. Weathering rates are determined in monthly steps for one annual scenario, thus reflecting seasonally changing weathering conditions, and then multiplied with the time step size.

Finally the diffusive hillslope transport is simulated.





The repetition of the sub-processes weathering, allochthonous supply, erosion, transport and accumulation leads to a growing regolith layer whose thickness in turn influences the weathering equations via the humped model: initially the weathering rate intensifies, from a certain thickness on it decreases again.

## 4. Results and evaluation

Results on thickness of regolith are available achieved via simulation of processes such as lithologically differentiated weathering of bedrock, erosion, transport and accumulation, as well as loess material supply from the last 50,000 years.

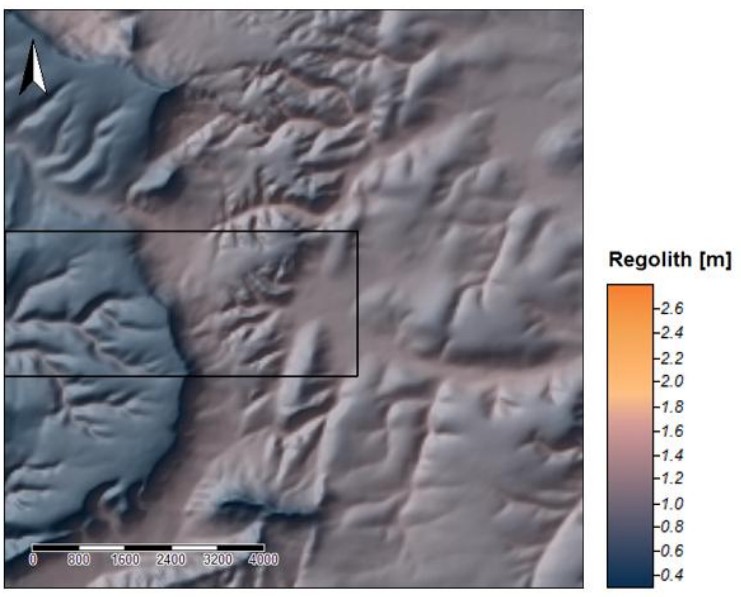

**Figure 7 First results of the SaLEM simulation in Ebergoetzen showing distributed regolith thicknesses resulting from 50 ka modeling. The rectangle indicates the area where a first validation of the results was conducted.**

These modelling data provide a picture of the spatial differentiation of regolith thickness for the study area (Fig. 7): valley areas are equipped with a massive filling up to several meters, whereas on ridges and nearby to steep slopes the thickness of the regolith tends towards zero. To the East of the study area the total thickness generally increases. Small tributary valleys have fillings thicker than the large main valley (in the center of the area), which drains to the east. Spatial differentiation within the slope areas clearly can be seen. This general picture is shown by all of the three variants the modeling was carried

out for: without initial sediment cover, with sediment cover of 50 centimeter thickness in general (Fig. 8) and finally with simulation of loess input according to accumulation rates (Fig. 9) proposed by Frechen et al. (2003). In detail the three variants differ significantly.



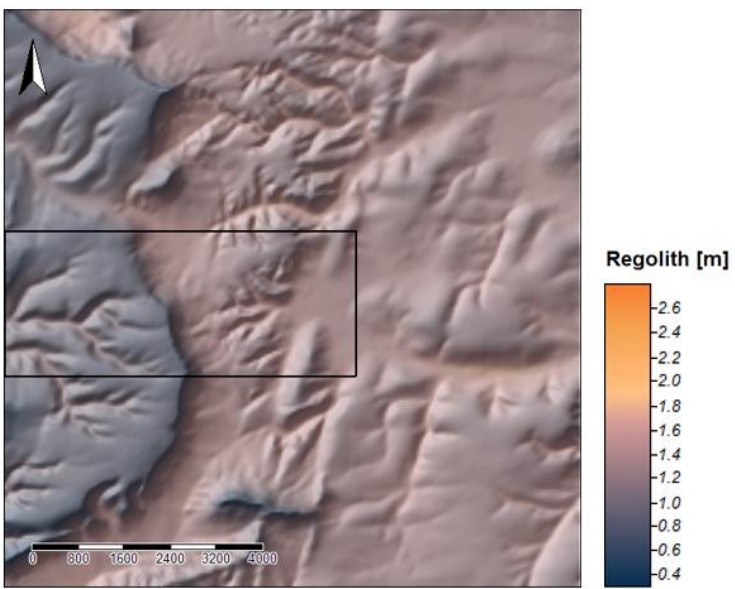

**Figure 8 First results of the SaLEM simulation in Ebergoetzen showing distributed regolith thicknesses resulting from 50 ka modeling with initial 50cm regolith cover. The rectangle indicates the area where a first validation of the results was conducted.**

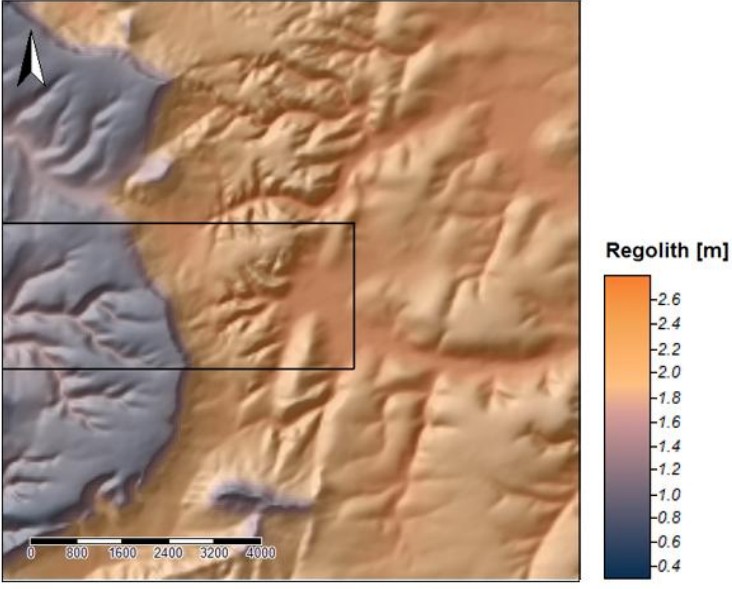

5  **Figure 9 First results of the SaLEM simulation in Ebergoetzen showing distributed regolith thicknesses resulting from 50 ka modeling including allochthonous input (loess). The rectangle indicates the area where a first validation of the results was conducted.**





Due to the lack of spatial data on properties of the regolith, which is next to the process understanding the motivation for our model development, the validation of the model results is not easy. There are no measurement data to validate it but hopefully will be collected in the near future (see future tasks). Legacy data in form of maps also do not exist, point measurements for other mapping projects (soil mapping campaigns) are only of limited use. Nevertheless, to give a first

impression, a compilation of available drilling point data from soil surveys is used to validate the trend of the results of the model regarding regolith thickness within a limited validation rectangle.

All available soil data for the area from the Lower Saxony State Office for Mining, Energy and Geology (LBEG) were collected (1141 point data within the validation rectangle, source: LBEG, soil profile data base, Fig. 10). However, since these are manually collected data for soil mapping projects, in most cases the total thickness of the regolith cover is not

completely recorded. Therefore, the depth of the weathered C-horizon was extracted for each profile although this value was set rather arbitrarily to 100cm for many locations due to the applied method (manual drilling) which cannot drill in deeper.

The depths of the C-horizons of the profiles were averaged for different process areas (separately for the stratigraphic units of the simplified Geological Map (Fig. 10, Elbracht, 2000) and for terrain positions of the simplified Geomorphographic map (Fig. 12, LBEG & scilands GmbH, 2008) and then compared with the generated model data, also averaged for the process

areas.

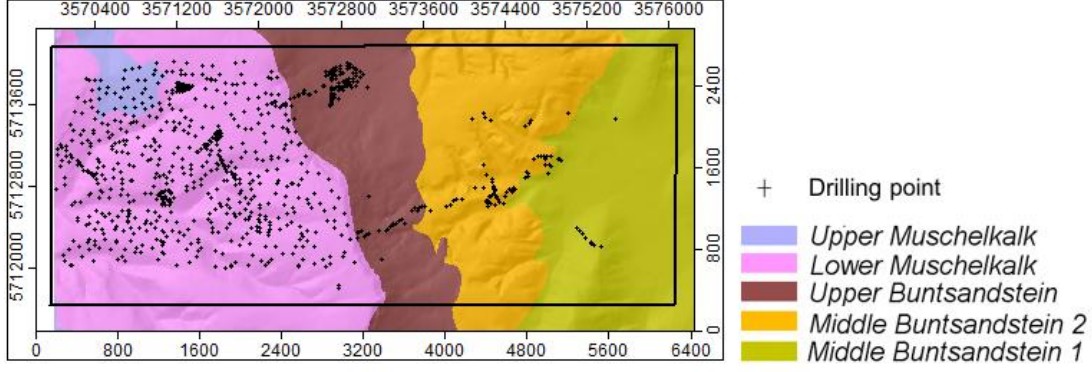

**Figure 10 Drilling points (n = 1141) from LBEG soil profile data base on the simplified geological units within the validation rectangle.**

The trend read in the profile data could be confirmed: in the process area, which is defined by the occurrence of the

stratigraphic unit of the Lower Muschelkalk limestone, the lowest average regolith thickness was modelled. For the three units of Buntsandstein on the other hand substantially higher mean thicknesses appeared. The modelled differentiation between Upper Buntsandstein, Middle Buntsandstein1 and Middle Buntsandstein2 could not be confirmed by the profile data because here the average values of all the units slightly fluctuate in a similar manner around the at least partly artificial maximum value of the profile depth. (Fig. 11).

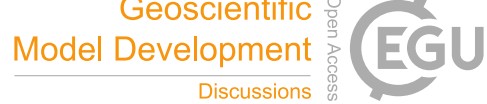



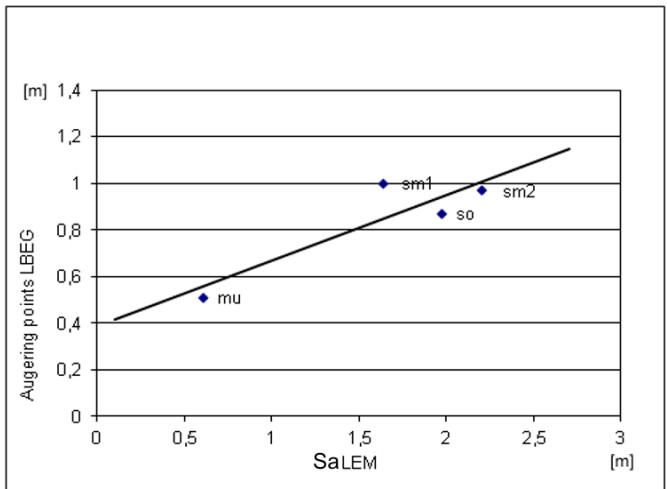

**Figure 11 The average thickness values [m] of the augering points compared to the average values of the SaLEM model run within the geological units of the validation rectangle. Lower Muschelkalk (mu), Upper Buntsandstein (so), Middle Buntsandstein2 (sm2), Middle Buntsandstein1 (sm1)**

5    For the hierarchically higher units of the geomorphographic map (bottom areas, slopes, summit areas stand for the relative bottom, middle and top, Fig. 12), also the trends in the profile data are reproduced in the model data: the lowest mean thicknesses was measured and also modelled in the summit areas of the terrain, higher mean thicknesses in slope and bottom area positions (Fig. 13). The fact that most profile data were set to an artificial depth of 100 cm is even more evident here: for slopes and bottom areas SaLEM clearly produces different average thicknesses, in profile data this difference is far less

10   obvious.

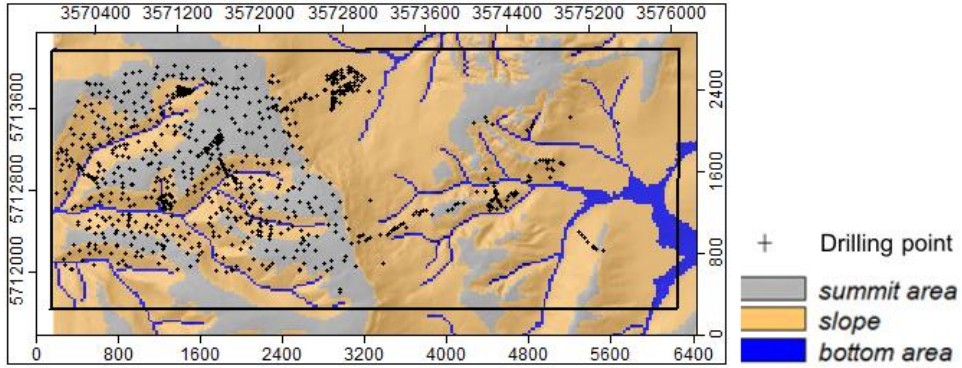

**Figure 12 Drilling points from LBEG soil profile data base on the units of the simplified geomorphographical map within the validation rectangle.**



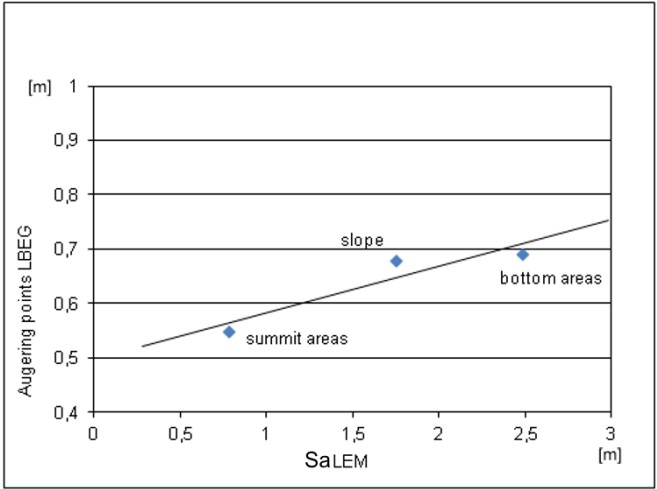

**Figure 13 The average regolith thickness values [m] of the drilling points compared to the average values of the SaLEM model run within the units of the geomorphographical map of the validation rectangle.**

The spatial differentiation of the model data within the individual process areas is not confirmed by the profile data. There are several possible reasons for this:

•       The spatial resolution of 50m grid cell size due to computing performance during the model development makes it impossible to reproduce the natural variability of regolith properties. Of course the natural variability is present in the measured data points instead.

•       The point data usually come with decimetres units, depths between full decimetres rarely occur. The focus is on the value of 100cm, which was set when the hand drill device could not reach the final depth of the profile.

•       The distribution of point data is not regular (Fig. 10). Approx. 74% of all points are located in the area of the Muschelkalk limestone, corresponding to a point density of about 32 points per km2, only 15% are in the area of the Upper Buntsandstein (point density approx 12points per km2), only 9% of the points fall into the Middle Buntsandstein2 area (point density 6 points per km2), in Middle Buntsandstein1 area only 1% (point density 0.6 points per km2 only). For the areas of the stratigraphic units of Buntsandstein no spatial differentiation corresponding to the grid size of the model is possible.

As a further result, the transport distances as well as the spatial distribution of the various rock types are assessed (Fig. 6), which is simulated by the tracer pathways. The kilometer-wide paths of Lower Muschelkalk and Upper Buntsandstein material are regarded as particularly plausible in the thicker regolith cover of the valleys. These data will soon be validated by means of deep drilling, but their evaluation is not yet available.

**5 Discussion and outlook**

The landscape evolution modelling approach (review article, see Tucker & Hancock, 2010) we introduce here is to create spatially differentiated modeling data of soil parent material properties. To make things clear it is not designed to explain the





shape of a landscape as universal and comprehensive as Perron et al. (2012) did. In this approach here we're looking into the recent past (and hopefully soon into the nearest future) to predict the properties of soil parent materials by simulating a set of processes involved.

Spatio-temporal modeling of these first-order processes of regolith-formation in SaLEM makes use of known physical relationships if possible. When there is no data available for calculation of process variables the modeling relies on parameterizations. For instance data of climate variables are used for weathering equations; the weathering resistance of different rock layers instead is parameterized by rank data from Gehrt (2008). Another example is the assumption about the spatial distribution of loess accumulation rate, which is composed out of a DEM-derived index and the in situ loess accumulation rate determined by Frechen et al. (2003). In later phases of expansion of the model, these parameterizations might be substituted by measured data or data from other sources.

A central problem of landscape evolution modeling is the lack of knowledge about the shape of the earth's surface in the geological past. An important part of the model starting situation cannot be derived: the topography of today's landscape may be the result of an infinite number of starting situations. A backward modeling therefore is excluded. In the case of modeling regolith properties, which is controlled by lithological, climatic and topographic conditions, the difference to the current appearance of the landscape for us is seen as less serious.

The process of regolith-evolution during the LGM is a complicated intermeshing of many different subprocesses. With SaLEM, initial results are obtained with certain validity. However SaLEM covers only a few sub-processes at this stage. We therefore have concrete ideas for the next steps:

In the near future we will strive for more realistic parameterization of the weathering properties of the lithological units using field (rock mass strength) and laboratory data (mineralogy). This aims to objectify the assessment of the lithologically differentiated weathering resistance. We will further modify the transport functions for different lithological materials and elaborate a suitable approach to dynamically model textural changes in the regolith-evolution. The latter is a challenge, especially for the computational implementation. We will lay emphasis on the calibration of the existing model parameters by considering the results of a deep drilling campaign conducted in 2012 and 2013. Unconsolidated fillings of valleys were sampled at different positions in the area. With these data we have an occasional glimpse into regolith development. Another focus of future research will be the creation of validation data basis. Recent developments of non-invasive geophysical measurements give hope that at least for some areas we can generate validation data to prove our modeling results in the future. To reflect the recognition that also suddenly occurring events affecting the evolution of regolith, we will incorporate existing models of discrete events (landslides, floods).

## 6 Code availability

The SAGA source code repository, including SaLEM version 1.0, is hosted at https://sourceforge.net/projects/saga-gis/ using a git repository. Read only access is possible without login. Alternatively, the source code and binaries can be downloaded





directly from the files section at https://sourceforge.net/projects/saga-gis/. SaLEM has been included here with SAGA version 6.0.0. Within the source code tree it is located at 'src/tools/simulation/sim_landscape_evolution'. The data for the test site used in this study can be downloaded from the files section too.

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
