# Peer review of "SaLEM (v1.0) - A Soil and Landscape Evolution Model for simulation of regolith depth in periglacial environments"

_Geoscientific Model Development, 2017_

## Editor Comment (EC1) · L. Gross (Editor) · 6 Nov 2017

Michael

GMD is encouraging authors to provide persistent access to the released source code and data through the use of a DOI which then can be cited in the paper. For projects in GitHub a DOI can be created using Zenodo, see https://guides.github.com/activities/citable-code/ for details. Please note that in the code accessibility section you can still point the reader to the GitHub repository for the newest version even if you use a DOI for the relevant release.

[Figure]

All the best

Lutz Gross GMD Executive Editor
* * *

---

## Referee Comment (RC1) · T. Coulthard (Referee) · 14 Nov 2017

The overall aim of the paper is good - and the contribution of a new/alternative way of integrating a soil development model, within a Landscape Evolution Model (LEM), within the SAGA GIS is great.

There are some issues that I think the paper would benefit greatly from being corrected. Mainly structure/writing/presentation but also some science. There are more detailed comments and corrections in the annotated PDF attached.

Firstly, the introduction is very hard to follow, and would benifit from a re-structuring.

Most of the main items, points and justifications are there, but could be far more clearly presented with a structure first outlining the background science, the research gap and how this model/paper will address that gap. This should also bring in background of existing models that look at Soil/LEM interaction (including that by Willgoose and co-workers, VanVallegan and Temme etc..).

The description of the model operation and its parameterisation is generally good and easy to follow. With the odd sentence that is hard to follow (identified in the PDF).

The validation section is rather downbeat $\sim$ this does beg the question is it worth developing a model for this area if you have no/so little data to compare it to? It's interesting to note that whilst the model predicts the pattern of soil development correctly (slope, peak, base etc..) it over estimates the regolith thickness considerably (by >3). I can't help but wonder if this is simply having too high rates in the weathering/generation of soil from the parent material - and this could be greatly improved by reducing these rates? In effect calibrating the model? I would suggest this as a good improvement for the model performance and that section of the paper.

Some of the arguments in the discussion/conclusion seem a bit out of place for a paper that is describing the development of a new model - these may be better saved for a different future publication that may look more at what the results mean (rather than how the results were made).

Please also note the supplement to this comment:
https://www.geosci-model-dev-discuss.net/gmd-2017-218/gmd-2017-218-RC1-supplement.pdf

**Supplement:**

[revised manuscript text omitted]

---

## Referee Comment (RC2) · Anonymous Referee #2 · 16 Nov 2017

This manuscript is dealing with a new concept for dynamic modeling of parent soil material and periglacial layers by the parameters of different bedrock material and climate conditions. This is an innovative approach to delineate soil texture in areas of higher latitude. However, as the authors mention themselves, it is a first step on the way to a complex model, which needs to include at least more calibration parameters and especially more precise validation data during further research. Actually, this kind of research meets the state of art for digital soil mapping and therefore meets the subject of the journal Geoscientific Model Development and should be published after some revisions.

[Figure]

A big difference of the presented model SaLEM to existing procedures is the dynamic modelling of the temporal development and vertical extension of the regolith layer by the parametrization of the geomorphological processes and paleoclimate. However, it is a common method to divide the periglacial layers in up to 4 stratigraphic units, also named cover-beds or in the German soil science known as Basislage, Mittellage, Hauptlage and sometimes Oberlage. The stratification of the periglacial layers is connected to periods of specific climatic conditions, which induce different morphodynamic processes. The variation in geomorphological activity should occur in the processed data when including paleoclimatic time sequences. In addition, the effective amount of aeolian sediment input during different glacial periods has an influence on the vertical composition of the regolith. This background makes it is necessary do discuss this phenomenon and how it is treated in model with few sentences. Is there any evidence for vertical soil texture distribution in the processed data? There is no information about this problem in the manuscript. In most cases, SaLEM produces regolith thickness of more than 1 m. In contrast, the maximum depth of soil cover in the validation data is 1 m. Therefore, the evaluation by this data makes only sense for mu or summit areas. Spending few days of fieldwork to execute some Pürckhauer drillings deeper than 1 m at specific sites would strongly increase the reliability of validation of the model results. At several parts of the manuscript dimensions, parameter and values need more detailed explication.

You will find detailed comments in the supplemented pdf.

Please also note the supplement to this comment:
https://www.geosci-model-dev-discuss.net/gmd-2017-218/gmd-2017-218-RC2-supplement.pdf

**Supplement:**

Review of the discussion paper in Geoscientific Model Development:

SaLEM (v1.0) – A Soil and Landscape Evolution Model for simulation of regolith in periglacial environments

This manuscript is dealing with a new concept for dynamic modeling of parent soil material and periglacial layers by the parameters of different bedrock material and climate conditions. This is an innovative approach to delineate soil texture in areas of higher latitude. However, as the authors mention themselves, it is a first step on the way to a complex model, which needs to include at least more calibration parameters and especially more precise validation data during further research. Actually, this kind of research meets the state of art for digital soil mapping and therefore meets the subject of the journal Geoscientific Model Development and should be published after small revisions.

In the following comments, I will first give some general ideas about adding specific information into the thematic frame of the manuscript. Then I propose some potential changes or supplements line by line in detail comments.

General comments:

A big difference of the presented model SaLEM to existing procedures is the dynamic modelling of the temporal development and vertical extension of the regolith layer by the parametrization of the geomorphological processes and paleoclimate. However, it is a common method to divide the periglacial layers in up to 4 stratigraphic units, also named cover-beds or in the German soil science known as Basislage, Mittellage, Hauptlage and sometimes Oberlage. The stratification of the periglacial layers is connected to periods of specific climatic conditions, which induce different morpho-dynamic processes. The variation in geomorphological activity should occur in the processed data when including paleoclimatic time sequences. In addition, the effective amount of aeolian sediment input during different glacial periods has an influence on the vertical composition of the regolith. This background makes it is necessary do discuss this phenomenon and how it is treated in model with few sentences. Is there any evidence for vertical soil texture distribution in the processed data? There is no information about this problem in the manuscript.

In most cases, SaLEM produces regolith thickness of more than 1 m. In contrast, the maximum depth of soil cover in the validation data is 1 m. Therefore, the evaluation by this data makes only sense for mu or summit areas. Spending few days of fieldwork to execute some Pürckhauer drillings deeper than 1 m at specific sites would strongly increase the reliability of validation of the model results.

At several parts of the manuscript dimensions, parameter and values need more detailed explication. This I will mention in the detailed comments.

Detailed comments:

Page Line: Comment

P2 L4:   authors (they are 2!)

P2 L5:   give detailed information about scale of time and area, see also repetition of this fact at page 6 line 5-8

P5 L16:  During the glacial periods …

P5 L23:  distinct thickness : If it is distinct give accurate amount of thickness!

P6 L16:  .. considered as highly evident ..

P6 L17:  . However, for the period ..

P6 L18:  .. for the initial topography

P 6 L19:  .. covering the bedrock, is the results of various natural processes ….

P6 L20:  Solid bedrock is weakened by ..

P6 L23:  "discrete periods": Please give detailed information about time and naming of the periods

P6 L24   .. material are evident, in particular aeolian sediments like loess.

P6 L25:  There are several multi-material-layers covering the solid rock ..

P6 L26:  "coat" change to "cover"

P6 L30:  .. not available for all …

P7 L3:   .. it became ..

P7 L5-6: Give literature or internet source for high resolution data

P7 L10:  "calibrated"; Calibration would mean more computing than only constantly increasing the values. I would prefer to say "adapted" or "transformed".

P7 L17:  What is the required temporal resolution of the model? Please give information.

P7 L19:  What is the amount of the temperature value increase for the study area? Please inform.

P7 L20:  Revise sentence, because you begin talking about the temperature signal, which has nothing to do with the spatial resolution of the precipitation data.

P9 L14: Erase point behind (2008)

P9 L20: Explain the parameters T, Tmin, Tmax, is it daily, annual, seasonal temperature? In the formula you use an "a" and in the text an "α" for the buffering parameter.

P12 L15: Describe the constitution of the initial regolith cover in detail. How it was designed?

P13 L14-17: This sentence about the three variants of modelling should appear in the beginning of the paragraph because it needs to be explained first that the result shown in figure 7 is computed without aeolian deposits.

P13 L14:  .. sediment cover (7), ..

P15 L14: I think you used the variation 3 of the generated model data for validation. This information is missing and need to be given here.

P17 L6, L10, L13: Space between number an m, cm or points

Figure 1c): Black lines in the map need to be explained in the legend, where is the mm, middle Muschelkalk?

Figure 2:    .. (dashed line) (after Alley, 2000) and the …

Figure 3a): Give more space for the coordinates in the map frame. The geographic coordinates in the map should be placed more systematically.
Figure 3b) and c) are inverted in the legend, the x-axes as time scale should be divided into 4 seasons

Figures 11, 13: Please indicate the statistical background in the diagrams, for example maximum and minimum values, standard deviation along x and y-axes.

---

## Short Comment (SC1) · 12 Dec 2017

Dear Lutz, thanks for your suggestion to create an DOI for SAGA and this particular version. The project is hosted at SourceForge and uses SF's GIT server, not GitHub. So there is no automation supported for generating DOI's each time a release is out. Anyway, we created one manually through Zenodo. Here it is: https://doi.org/10.5281/zenodo.1063915 Best wishes, Olaf

---

## Author Comment (AC1) · 13 Dec 2017

Dear Tom Coulthard,

thank you very much for your important comments and suggestions. I've incorporated these into the manuscript as best I could. Attached is a list of my answers to all corrections and comments.

It was a pleasure to me to raise the level of this paper with your help. Best regards Michael

Please also note the supplement to this comment:
https://www.geosci-model-dev-discuss.net/gmd-2017-218/gmd-2017-218-AC1-supplement.pdf

––––––––––––––––––––––––––––––

**[GMDD](https://gmd.copernicus.org)**
[Figure]

**Supplement:**

Technical corrections Review1:

P1-L19: "present day" instead of "nowadays"

P1-L19: "Particular" deleted

P1-L20: "were" instead of "ruled"

P1-L21: "be substantially improved" instead of "improve substantially"

P1-L21: "would" instead of "were"

P1-L24: "to model "instead of "for the modelling of"

P2-L9: "essential" deleted

P2-L10: "20th century" instead of "last century"

P15-L1: ", which is next to the process understanding the motivation for our model development" deleted

P15-L6-8: "There are no measurement data to validate it but hopefully will be collected in the near future (see future tasks). Legacy data in form of maps also do not exist, point measurements for other mapping projects (soil mapping campaigns) are only of limited use. Nevertheless," deleted

P18-L2: "(and hopefully soon into the nearest future)" deleted

Your comments in the supplement of review 1:

*Sticky note R1"why are you introducing this tool then?to answer this question? This needs to be made much clearer"*
AR: should be clear now after restructuring

*Sticky note R1: "why is it only applicable to this area? Is this a suggestion of the models weakness or a lack of confidence? Its not really clear why this is said here."*
AR: the sentence in question was deleted

*Sticky note R1: "OK - but you've already said that your model is site specific! so...."*
AR:

*Sticky note R1: "this sentence is weak and needs clarification - what do you mean by deductive models here? As in induction/vs deduction or a different definition?"*
AR: Exactly, it's meant as in induction/deduction. In deductive models according to Boehner (2006) dynamic processes are represented by physical laws resp. physical analogies whereas inductive models point out relations by statistically analyzing empirical data.
I provide the translation of a figure of one of Boehners publications. Here he uses empirical (for inductive) and numerical (instead of deductive).

[Figure]

*Sticky note R1: "in what sense? Clearly define what the model aims to solve"*
AR: See next one

*Sticky note R1: "as a general comment on the introduction: All the main parts are there - but the order and structure needs work. There is a mixture of descriptions of previous work and processes with descriptions of what the model will do. This would be much clearer if the issue/problem to be adressed were clearly described - then looking at previous work/models that have looked at pedogenesis in LEM's - and then on to why there is a niche/gap for this model. Then finally what you plan to do in the paper etc..*
AR: I restructured the introduction according to your suggestions and added some recent works to complete the framework of soil-landscape models I mention here.

*Sticky note R1: "This sentence needs re-working its not clear whether you are talkign about world records, or global data.. also needs a reference."*
AR: I limited it to the modeling of global paleo data and added Kageyama (2016) as a reference.

*Sticky note R1: So salem is over predicting regolith depth by c.3m? Could this not be calibrated out by altering parameters in the rock>soil components? It would appear that the relationship between summit, slope and bottom is correct (e.g. the dynamics of the model are correct) but the amounts are wrong...*
AR: In my opinion the model prediction of estimated values up to 3m in valley floors is plausible. Only the validation data does not reflect this due to the limitations of that data source. No one has drilled manually deep enough that he reached the bedrock. In the meantime a drilling campaign with heavy equipment was carried out by the federal state agency LBEG. The results showed even deeper values (1.5m to 13.4m) in the valley floors. This new data source contains ten boreholes in the site Ebergoetzen, but only two in the validation rectangle. To avoid confusion I did not mix this data with the manually drilled data source I analyzed in this paper. The new data source is publically available so you may have a look at https://nibis.lbeg.de/cardomap3/?lang=en# and type in one of these numbers into the search window of the site: 4426GE0049, 4426GE0050, 4426GE0051, 4426GE0055, 4426GE0056, 426GE0059, 4426GE0061, 4426GE0062, 4426GE0103, 4426GE0104. When you click "further information" you get the depth values of each.
My idea is to carry on with SaLEM by calibrating the model especially the composition of the unconsolidated layer reflected in the tracer by means of this new data source.

*Sticky note R1: what did Perron do/say? You need to explain!*

AR: Added half of a sentence to make clear that Perron follows a more comprehensive approach of landscape evolution simulating the branches of river networks.

*Sticky note R1: I'm not sure what this section adds to the paper... I would consider removing it unless the point is a central one of the model reporting (which I don;t think it is...?)*
AR: Removed

---

## Author Comment (AC2) · 14 Dec 2017

Dear anonymous referee #2,

thank you very much for your comments and suggestions, . I've incorporated these into the manuscript as best as I could.

To answer your comments first: It is definitely right to say that SaLEM "is a first step on the way to a complex model, which needs to include at least more calibration parameters and especially more precise validation data during further research." That's want we want to do in the future! The 'problem' of SaLEM predicting regolith depth values

more than 1m which is not reflected by the validation data is starting to get slightly bet-
ter: In the meantime a drilling campaign with heavy equipment was carried out by the
federal state agency LBEG. The results showed even deeper values (1.5m to 13.4m) in
the valley floors. This new data source contains ten boreholes in the site Ebergoetzen,
but only two in the validation rectangle. To avoid confusion I did not mix this data with
the manually drilled data source I analyzed in this paper. The new data source is pub-
lically available so you may have a look at https://nibis.lbeg.de/cardomap3/?lang=en#
and type in one of these numbers into the search window of the website: 4426GE0049,
4426GE0050, 4426GE0051, 4426GE0055, 4426GE0056, 426GE0059, 4426GE0061,
4426GE0062, 4426GE0103, 4426GE0104. When you click "further information" you
get the depth values of each. You'll see that the SaLEM prediction of valley filling was
really conservative!

We are aware of the sequences of different times of loess input and reworking and
mixing this allochthonous material with weathered in situ material from the bedrock.
We also have in mind the classification concept of four different stratigraphic layers of
periglacial material you mentioned. This concept is not part of the modeling via SaLEM.
When SaLEM will be able to predict different layers of material composition it will be
time to proof whether the model can meet the differentiation known as 'Deckschichten'.
At the moment this is too early. But we're working on that: the tracer concept we
introduced here puts us in the position to follow the pathways of material and derive
the prediction of material composition for every grid cell. So I'm quite sure that we can
reach that goal soon!

Thanks again for the time and effort you spend.

Michael

Please also note the supplement to this comment:
https://www.geosci-model-dev-discuss.net/gmd-2017-218/gmd-2017-218-AC2-
supplement.pdf

[Figure]

**Supplement:**

Technical corrections Review#2:

P2 L4: authors (they are 2!) –
AR: corrected

P2 L5: give detailed information about scale of time and area, see also repetition of this fact at page 6 line 5-8 –
AR: over longer geologic time periods (added "several Ma") for large areas (added "thousands of square kilometers").

P5 L16: During the glacial periods …
AR: corrected

P5 L23: distinct thickness : If it is distinct give accurate amount of thickness!
AR: changed to "different but considerable thickness".

P6 L16: .. considered as highly evident ..
AR: changed

P6 L17: . However, for the period ..
AR: corrected

P6 L18: .. for the initial topography
AR: corrected

P 6 L19: .. covering the bedrock, is the results of various natural processes .…
AR: changed

P6 L20: Solid bedrock is weakened by ..
AR: corrected

P6 L23: "discrete periods": Please give detailed information about time and naming of the periods
AR: I changed "Pleistocene" to late Pleistocene and added this information: "during the Middle and Upper Weichselian when aeolian sediments like loess were accumulated and reworked with autochthonous material (Frechen 2003)."

P6 L24 .. material are evident, in particular aeolian sediments like loess.
AR: changed

P6 L25: There are several multi-material-layers covering the solid rock ..
AR: I replaced "a" with "several"

P6 L26: "coat" change to "cover"
AR. Changed

P6 L30: .. not available for all …
AR: changed

P7 L3: .. it became ..

AR: corrected

P7 L5-6: Give literature or internet source for high resolution data
AR: I changed "Palaeo-climate modeling data of world records" to "Palaeo climate modeling of global data" and cited Kageyama, 2016

P7 L10: "calibrated"; Calibration would mean more computing than only constantly increasing the values. I would prefer to say "adapted" or "transformed".
AR: decided to use „adapted"

P7 L17: What is the required temporal resolution of the model? Please give information.
AR: It is conducted with a time step of 100 years at the moment. But it is a flexible parameter for users with different data for climate input.

P7 L19: What is the amount of the temperature value increase for the study area? Please inform.
AR: The curve was adapted to meet the annual mean temperature value of 8 degrees at the testsite "Ebergoetzen"

P7 L20: Revise sentence, because you begin talking about the temperature signal, which has nothing to do with the spatial resolution of the precipitation data.
AR: "and precipitation" was deleted

P9 L14: Erase point behind (2008)
AR: OK

P9 L20: Explain the parameters T, Tmin, Tmax, is it daily, annual, seasonal temperature? In the formula you use an "a" and in the text an "α" for the buffering parameter.
AR: added "T is the Mean Annual Average Temperature (MAAT) in °C, Tmax is the maximum MAAT in °C, Tmin is the minimum MAAT in °C within the time step", the a in the formula was replaced by "α"

P12 L15: Describe the constitution of the initial regolith cover in detail. How it was designed?
AR: Here I describe only the initial depth of the regolith cover. The user can decide for a general value (5cm, 10cm,…) or include a grid with spatially distributed depth values. I replaced "definition" with "depth".

P13 L14-17: This sentence about the three variants of modelling should appear in the beginning of the paragraph because it needs to be explained first that the result shown in figure 7 is computed without aeolian deposits.
AR: I shifted the sentences so that the paragraphs starts with a description of the variants.

P13 L14: .. sediment cover (7), ..
AR: not sure what is meant…

P15 L14: I think you used the variation 3 of the generated model data for validation. This information is missing and need to be given here.
AR: You're right. The missing information was added!

P17 L6, L10, L13: Space between number an m, cm or points
AR: Corrected!

Figure 1c): Black lines in the map need to be explained in the legend, where is the mm, middle Muschelkalk?
AR I completed the legend of figure 1c). The Middle Muschelkalk fell victim to the simplification of the Geological data when the transformation to the geological model was conducted. There were only very small areas anyway.

Figure 2: .. (dashed line) (after Alley, 2000) and the …
AR: corrected

Figure 3a): Give more space for the coordinates in the map frame. The geographic coordinates in the map should be placed more systematically.
AR: I decided to produce the map without map frame with coordinates. Instead the map now has a scale bar and systematic annotation of the graticule.

Figure 3b) and c) are inverted in the legend, the x-axes as time scale should be divided into 4 seasons
AR: Corrected. I introduced a new partitioning of the x axis.

Figures 11, 13: Please indicate the statistical background in the diagrams, for example maximum and minimum values, standard deviation along x and y-axes.
AR: I produced the figures 11 and 13 again with arrows that indicate the standard deviation for both the augering points from LBEG and the modeling results. Min / max values seemed to me not very meaningful in this case.

---

## Referee Report (RR1)

Review of the revised manuscript in Geoscientific Model Development:

SaLEM (v1.0) – A Soil and Landscape Evolution Model for simulation of regolith in periglacial environments

This manuscript is dealing with a new concept for dynamic modeling of parent soil material and periglacial layers by the parameters of different bedrock material and climate conditions. This is the first step of an innovative modelling approach to delineate soil texture in areas of higher latitude. This research represents the state of art for digital soil mapping and therefore meets the subject of the journal Geoscientific Model Development and needs to be published. In my opinion, the scientific output is well and the manuscript has in an accurate structure and adequate content. However, after carefully reading the manuscript I found some small mistakes and several sentences, which should be rearranged. That's all.

Detailed comments:

Page Line: Comment

| | |
|---|---|
| P1 L30 | composition, porosity |
| P2 L6-L9 | Very long sentence, please make 2 of it |
| P 3 L11 | Pilot project, yes but pilot area ? |
| Figure 1: | .. according to Ehlbrecht (2000) |
| P 5 L16 | .. which consists of sequences .. |
| P5 L17-18 | No complete sentence: Partly as insular very thick resources can be found |
| P6 L16 | .., which is considered .. |
| P6 L28 | (Frechen et al. 2003) |
| P7 L22 | The average period shown |
| Figure 3: | Kalnay et al. (1996) |
| P9 L6 | 50 m |
| Figure 4: | .. from Ehlbrecht (2000) |
| P13 L11 | Sentence: Results on thickness of regolith are available achieved via simulation of processes …(?) |
| Figure 9: | .. showing the distribution of regolith .. |
| P15 L5 | Maybe better ?:  .. the validation of the model results is challenging. |
| P15 L12 | 100 cm |
| P15 L12 | Maybe better ?: .. which cannot drill deeper into the soil. |
| P15 L13 | .. process areas separately .. |
| P18 L31 | Maybe better ?: .. spatial pattern of loess accumulation rate, .. |